# Relationship Between Swallowing Function and Low Serum Albumin Levels in Older Japanese People Aged ≥ 75 Years

**DOI:** 10.3390/healthcare12212197

**Published:** 2024-11-04

**Authors:** Komei Iwai, Tetsuji Azuma, Takatoshi Yonenaga, Yasuyuki Sasai, Yoshinari Komatsu, Koichiro Tabata, Taketsugu Nomura, Iwane Sugiura, Yujo Inagawa, Yusuke Matsumoto, Seiji Nakashima, Yoshikazu Abe, Takaaki Tomofuji

**Affiliations:** 1Department of Community Oral Health, School of Dentistry, Asahi University, 1-1851 Hozumi, Mizuho 501-0296, Gifu, Japan; tetsuji@dent.asahi-u.ac.jp (T.A.); yone0730@dent.asahi-u.ac.jp (T.Y.); ysssi0718@dent.asahi-u.ac.jp (Y.S.); km2-km2@dent.asahi-u.ac.jp (Y.K.); tabata-k@dent.asahi-u.ac.jp (K.T.); tomofu@dent.asahi-u.ac.jp (T.T.); 2Gifu Dental Association, 1-18 Minamidori, Kano-cho, Gifu 500-8486, Gifu, Japan; takezou@ma.ctk.ne.jp (T.N.); white@rhythm.ocn.ne.jp (I.S.); yujo@alato.ne.jp (Y.I.); whitebear@crest.ocn.ne.jp (Y.M.); semsynakashi@shopping3.gmobb.jp (S.N.); kjkcg552@ybb.ne.jp (Y.A.)

**Keywords:** cross-sectional study, older people, serum albumin level, swallowing function

## Abstract

Background/Objectives: This cross-sectional study aimed to investigate the relationship between swallowing function and low serum albumin levels in older Japanese people (aged ≥ 75 years). Methods: A total of 3258 participants (1325 males, 1933 females; mean age, 80.8 years) who had a dental checkup in Gifu City, Japan, between April 2020 and March 2021 were recruited. Swallowing function was assessed using the repetitive saliva swallow test, with poor swallowing function defined as swallowing fewer than three times in 30 s. A serum albumin level < 3.6 G/dL was considered low. Results: In total, 78 participants (2%) had a low serum albumin level. Furthermore, after adjusting for age, gender, circulatory disease, support/care-need certification, number of present teeth, and tongue and lip function, a low serum albumin level was positively associated with older (odds ratio [OR]: 1.115; 95% confidence interval [CI]: 1.064–1.169), male (OR: 2.208; 95% CI: 1.360–3.584), circulatory disease (OR: 1.829; 95% CI: 1.079–3.099), support/care-need certification (OR: 2.087; 95% CI: 1.208–3.606), and poor swallowing function (OR: 2.379; CI: 1.377–4.112). Conclusions: These results indicate that poor swallowing function was associated with a low serum albumin level in older Japanese people aged ≥ 75 years.

## 1. Introduction

The population is aging more rapidly in Japan than in any other country [1], and by 2024, approximately 20.2 million people, accounting for about 16% of the total population in Japan, will be aged ≥ 75 years [2]. One of the most important issues in super-aging societies is the presence of health problems resulting from poor nutritional status in older people [3,4]. Older people with poor nutritional status tend to have reduced muscle strength and bone mass, which increases the risk of not only falls and fractures but also falls resulting from a frail condition requiring care-need certification [5,6,7]. People with care-need certification tend to have lower energy consumption and food intake, which, in turn, leads to further poor nutritional status, resulting in a vicious cycle [8]. In Japan, the proportion of older people with such poor nutritional status is increasing [9]. Therefore, to help prevent older adults from requiring care-need certification, it is of great public health importance to investigate the factors associated with poor nutritional status.

Oral health is important because it not only helps maintain healthy teeth and gingiva but also contributes to nutritional status. Several studies have focused on the relationship between swallowing function and nutritional status. For example, it has been reported that dysphagia due to sarcopenia contributes to eating disorders and progresses to frailty because of poor nutritional status [10,11]. Furthermore, a study of inpatients in general hospitals found that the need for parenteral nutrition due to progressive dysphagia is associated with an increased risk of poor nutritional status [12]. These findings indicate that swallowing function is a potential factor associated with poor nutritional status; however, the strength of this association remains unclear. Therefore, clarifying this relationship in an epidemiological study may help facilitate the development of improved methods for the prevention of poor nutritional status.

Body mass index (BMI), weight change, and blood tests are currently used to diagnose poor nutritional status [13,14]. Of these tests, a serum albumin level < 3.6 G/dL is considered to indicate poor nutritional status, and such patients are required to participate in a health guidance program to improve their nutritional status in health checkups [15,16]. Therefore, evaluating serum albumin levels could be useful for clarifying the relationship between swallowing function and the risk of poor nutritional status.

Previous studies have reported an association between oral health and serum albumin levels. For example, an association has been found between periodontal condition and low serum albumin levels [17]. Furthermore, caries-prone individuals have been shown to have decreased serum albumin levels [18]. However, these reports investigated associations between specific oral diseases and serum albumin levels, so the community-level relationship between swallowing function and low serum albumin levels in older people remains unclear.

The National Database of Health Insurance of Japan (NDB) contains data on the serum albumin levels of community residents. In addition, in Gifu City, a community dental checkup is conducted once a year for older people ≥ 75 years, and the results of this checkup, including swallowing function, are stored in a database. Therefore, it is possible to analyze the relationship between swallowing function and low serum albumin levels by combining these data.

Given this background, in the present study, with the hypothesis that poor swallowing function may be associated with low serum albumin levels, we conducted a cross-sectional study to investigate the relationship between swallowing function and low serum albumin levels in older Japanese people ≥ 75 years.

## 2. Materials and Methods

### 2.1. Research Ethics

Our study was approved by the Ethics Committee of Asahi University (No. 33006; Approved 20 March 2020) and conducted in accordance with the Declaration of Helsinki. Written informed consent to use community dental checkup data was obtained from all participants. No informed consent was obtained for the NDB data because these data are anonymized.

### 2.2. Participants

Data were analyzed from local residents who had a community dental checkup in Gifu City, Gifu Prefecture, Japan. Exclusion criteria were people aged < 75 years and non-Japanese nationality, and all other people were considered as inclusion criteria. As a result, between April 2020 and March 2021, a total of 3515 older Japanese people aged ≥ 75 years were recruited to participate in our survey. Among these individuals, those with a medical history of dementia (*n* = 257) were excluded because they were considered likely to be unable to understand fully the instructions for tongue and lip or swallowing function testing [8]. Therefore, data from 3258 community-dwelling residents (1325 males, 1933 females; mean age, 80.8 years) were analyzed in our study (Figure 1).

### 2.3. Assessment of Swallowing Function

To assess swallowing function, we used the repetitive saliva swallowing test (RSST). Participants who swallowed fewer than three times in 30 s in the RSST were evaluated as having poor swallowing function [19].

### 2.4. Assessment of Serum Albumin Level

Information about serum albumin levels was collected from the NDB. In Japan, a serum albumin level < 3.6 G/dL is considered to indicate poor nutritional status, and patients with a serum albumin level < 3.6 G/dL are required to participate in a health guidance program at a health checkup to improve their nutritional status [15,16,20]. Therefore, in the present study, a serum albumin level < 3.6 G/dL was judged to be low [21]. Furthermore, three different categories of serum albumin level (<3.6 G/dL, ≥3.6 G/dL; <3.9 G/dL; and ≥3.9 G/dL) were used because, in our country, serum albumin levels ≥3.9 G/dL are considered normal [22].

### 2.5. Other Factors Collected from the NDB

Data regarding the participants’ age, gender, BMI, exercise habits, and presence of a malignant tumor, diabetes, dyslipidemia, musculoskeletal disorders, circulatory disease, support/care-need certification, and regular medications were also obtained from the NDB. We selected factors that may affect serum albumin levels [23,24,25,26,27,28,29,30]. Exercise habits were self-reported in a self-administered questionnaire. The questionnaire items regarding exercise habits included “I walk or perform an equivalent physical activity in daily life for at least an hour/day (presence or absence)”; participants who answered “presence” were defined as having a regular exercise habit [31,32].

### 2.6. Other Oral Factors

Data on the following oral factors were also obtained: number of present teeth, periodontal pocket depth, chewing function, and tongue and lip function. The Community Periodontal Index coded values were used to evaluate periodontal pocket depth ≥ 4 mm, with codes 1 and 2 being evaluated as a periodontal pocket depth ≥ 4 mm [33,34]. We also conducted the oral diadochokinesis test to assess tongue and lip function, with “poor tongue and lip function” defined as uttering fewer than 30 syllables (“Pa”, “Ta”, or “Ka”) in 5 s [35].

### 2.7. Statistical Analysis

The normality of the data was confirmed using D’Agostino’s K-squared test. Because all continuous variables were not normally distributed, data were expressed as medians (first and third quartiles). We used the chi-square test for categorical variables and the Mann–Whitney U test for continuous variables. Univariate and multivariate logistic regression analyses were performed with a serum albumin level < 3.6 G/dL or not as the dependent variable. Furthermore, on multivariate stepwise logistic regression analysis, variables with a *p*-value > 0.05 were excluded from the model. In addition, variables that were significantly different on univariate logistic regression analysis, in addition to gender and age, were selected as adjustment factors. The suitability of the model was confirmed by the Hosmer–Lemeshow test. The Hosmer–Lemeshow test is used to examine the goodness-of-fit of multivariate logistic regression analysis models, testing whether the observed event rates in a subgroup model fit the expected event rates. The Hosmer–Lemeshow test is considered to indicate a good fit with *p*-values > 0.05 [36]. All data were analyzed using SPSS statistical analysis software (SPSS Statistics, version 27; IBM Japan, Tokyo, Japan). All *p*-values ≤ 0.05 were considered significant.

## 3. Results

Table 1 shows the characteristics of the participants with a serum albumin level < 3.6 G/dL or not. In this study, 78 participants (2%) were defined as having a low serum albumin level and 399 (12%) as having poor swallowing function. Participants with a serum albumin level < 3.6 G/dL were more likely to be older (*p* < 0.001) and male (*p* = 0.030), as well as to have circulatory disease (*p* < 0.001), support/care-need certification (*p* < 0.001), poor tongue and lip function (*p* < 0.001), poor swallowing function (*p* < 0.001), and number of present teeth ≥ 20 (*p* = 0.001).

Table 2 shows the results of a comparison of serum albumin levels and swallowing function. Among the participants, 78, 339, and 2841 had a serum albumin level < 3.6 G/dL, ≥3.6 G/dL; <3.9 G/dL; and ≥3.9 G/dL, respectively. Furthermore, the proportions of these participants with poor swallowing function were 31%, 15%, and 11%, respectively, and these values decreased significantly with increasing serum albumin levels.

The results of the univariate logistic regression analysis with low serum albumin level as the dependent variable are shown in Table 3. A low serum albumin level was significantly associated with older age (odds ratio [OR]: 1.169; 95% confidence interval [CI]: 1.124–1.216), male gender (OR: 1.636; 95% CI: 1.043–2.566), circulatory disease (OR: 2.525; 95% CI: 1.512–4.217), support/care-need certification (OR: 4.220; 95% CI: 2.659–6.699), number of present teeth < 19 teeth (OR: 2.052; 95% CI: 1.308–3.218), poor tongue and lip function (OR: 2.278; 95% CI: 1.452–3.574), and poor swallowing function (OR: 3.324; 95% CI: 2.031–5.442).

Table 4 shows the adjusted OR and 95% CI for a low serum albumin level according to the analyzed factors in the participants. A low serum albumin level was positively associated with older age (OR: 1.115; 95% CI: 1.064–1.169), male gender (OR: 2.208; 95% CI: 1.360–3.584), circulatory disease (OR: 1.829; 95% CI: 1.079–3.099), support/care-need certification (OR: 2.087; 95% CI: 1.208–3.606), and poor swallowing function (OR: 2.379; 95% CI: 1.377–4.112) after adjusting for age, gender, circulatory disease, support/care-need certification, number of present teeth, tongue and lip function, and swallowing function.

## 4. Discussion

To the best of our knowledge, this is the first study to examine the association between swallowing function and low serum albumin levels among older Japanese people aged ≥ 75 years using data from the NDB. The results show that more participants with a serum albumin level < 3.6 G/dL had poor swallowing function compared with those with a serum albumin level ≥ 3.6 G/dL. Furthermore, participants with poor swallowing function had a lower serum albumin level than those with good swallowing function. Our analyses also revealed that a low serum albumin level was associated with poor swallowing function after adjusting for age, gender, circulatory disease, support/care-need certification, number of present teeth, and tongue and lip function. These observations suggest that poor swallowing function may be a risk factor for a low serum albumin level in older Japanese people aged ≥ 75 years.

Some possible mechanisms may explain the relationship between poor swallowing function and low serum albumin levels. Poor swallowing function induces difficulties with food intake [37,38], which may result in a decrease in the amount of food consumed, leading to poor nutritional status and low serum albumin levels. In addition, poor swallowing function may be associated with reduced dietary diversity [39]. A previous report found that poor swallowing function led to worse dietary diversity and interfered with the intake of vitamins and magnesium [40]. Vitamin and magnesium deficiencies are known to be associated with low serum albumin levels because they may be accompanied by hypocalcemia [41]. Therefore, participants with poor swallowing function may have had unbalanced eating habits and therefore been more likely to consume soft foods with high fat content, leading to inadequate nutrient intake and a low serum albumin level.

In the present study, three different categories of serum albumin level were used: <3.6 G/dL, ≥3.6 G/dL; <3.9 G/dL; and ≥3.9 G/dL. This is because, according to the Nutrition Improvement Manual of the Ministry of Health, Labour and Welfare of Japan, a serum albumin level ≥ 3.9 G/dL is considered normal [42]. In our study, as serum albumin levels approached normal, the proportion of participants with poor swallowing function decreased significantly. In other words, the prevention of poor swallowing function may be associated with not only the prevention of poor nutritional status but also the maintenance of normal nutritional status.

In this study, the RSST was used to assess swallowing function. Defining participants who swallowed fewer than three times in 30 s in the RSST as having poor swallowing function has been widely used as an excellent method of assessing individual swallowing function [19,43,44]. On the other hand, a previous study reported that the RSST had a sensitivity of 98% and a specificity of 66% [45]. Therefore, it is possible that some of the participants in the present study may have had a false-positive result, and this remains an issue for future research.

Previous studies have reported finding a relationship between swallowing function and systemic health. For example, dysphagia, as assessed using the 10-item self-administered Eating Assessment Tool, has been shown to be associated with the development of diabetes [46]. It has also been reported that poor swallowing function is associated with the development of dementia and increased future mortality [47,48]. These assessments of swallowing function include self-administered questionnaires, the RSST, and patients hospitalized for impaired swallowing function. It has also been reported that dysphagia is a major risk factor for aspiration pneumonia [49]. A systematic review also reported finding an association between poor swallowing function and hypertension [50]. Therefore, the findings of the present and previous studies support the notion that poor swallowing function may be detrimental to systemic health.

In the present study, no association was found between number of present teeth and low serum albumin levels; however, a previous study reported finding such an association [51]. This discrepancy may be related to the difference between studies in the proportion of participants with a number of present teeth ≥ 20. In the previous study, the proportion of participants with a number of present teeth ≥ 20 was about 50% [51], compared with about 67% in the present study. Furthermore, in the Survey of Dental Diseases in 2022, the proportion of individuals aged 80 years, which is the same average age as in the present study, who had a number of present teeth ≥ 20 was about 51% [52]. In other words, these findings may have been influenced by the fact that the participants in the present study were more dentally literate than the general population in Japan.

On the other hand, in the present study, a trend toward an association between tongue and lip function and low serum albumin levels was seen in the univariate analysis; however, no significant association was seen in the multivariate analysis. A relationship has been reported between tongue and lip function and serum albumin levels as diagnosed using an oral diadochokinesis test, as in the present study [53]. However, the target age group in that study was younger than that in the present study (≥65 years vs. ≥75 years, respectively), and swallowing function was not taken into account in multivariate analysis. Therefore, the results may have differed because the age range of the participants in the present study was expanded. In addition, if tongue and lip function and swallowing function were simultaneously considered as factors associated with low serum albumin levels, swallowing function could be more relevant, as in the present study.

In this study, a low serum albumin level was associated with age, circulatory disease, and support/care-need certification. According to the National Health and Nutrition Survey in 2019, the proportion of older people in Japan with poor nutrition as assessed by BMI has increased with increasing age [9]. In addition, albumin is an essential protein that binds and transports various drugs and substances to maintain venous oncotic pressure and influences the circulatory system, and a decreased albumin level has been shown to serve as a strong predictor of an increased risk for cardiovascular disease [54]. Furthermore, compared with healthy people, older people with support/care-need certification have been shown to have worse nutrition status as assessed by BMI, with rates in Japan ranging from 20 to 40% [54]. With respect to these factors, the results of the present study, which used serum albumin levels to assess nutrition status, are similar to those of previous studies that used BMI.

In Japan, a serum albumin level < 3.6 G/dL is considered to indicate poor nutrition [15,16]. In addition, poor nutrition in older people is known to be closely associated with an increased risk of not only frailty but also easy infection and death [55]. Therefore, the clinical and theoretical implication of the present study is that dentists routinely check the swallowing function of the elderly and that preventing poor swallowing function may be associated with a decreased risk of easy infection and death due to poor nutritional status among the elderly. In other words, our study suggests that regular swallowing function checks by dentists are important to prevent poor nutritional status.

However, this study has some limitations. First, the participants may have been a more health-conscious sample than the general population because they voluntarily attended a community dental checkup. Especially, in our study, the proportion of participants with serum albumin levels < 3.6 G/dL was 2.3% (78 participants). According to the National Health and Nutrition Examination Survey in 2022, the proportion of older Japanese people ≥ 75 years with poor nutritional status was about 15%, which is higher than the results of our study [9]. Therefore, the results may differ when targeting samples with different health conditions. Second, as this study had a cross-sectional design, we could not examine causal relationships. Additional longitudinal studies are needed to investigate the relationship between swallowing function and low serum albumin levels. Third, while serum albumin level is an indicator of nutritional status, it is also used as an indicator of impaired liver function [56]. Therefore, participants with impaired liver function in our study may show false positive results. Finally, the presence or absence of various health conditions/indicators (e.g., osteoporosis, fracture, daily calorie intake) was unclear because these data are not contained in the NDB.

## 5. Conclusions

The results of this study indicate that poor swallowing function is associated with a low serum albumin level in older Japanese people aged ≥ 75 years. Therefore, assessing swallowing function using the RSST at medical or dental checkups could help screen for the presence or absence of poor nutritional status.

## Figures and Tables

**Figure 1 healthcare-12-02197-f001:**
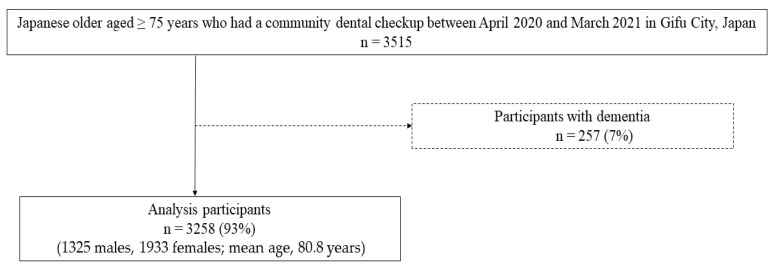
Flowchart of our study.

**Table 1 healthcare-12-02197-t001:** Characteristics of the participants with a serum albumin level < 3.6 g/dL or not.

Factor	Serum Albumin Level		*p-*Value
	3.6 G/dL≤ (*n* = 3180)	<3.6 G/dL (*n* = 78)	
Age (years)	80 (77, 83)	85 (80, 88)	<0.001 *
Gender ^a^	1284 (40%)	41 (52%)	0.030
BMI (a: kg/m^2^)			
25.0 ≤ a	659 (21%)	10 (12%)	0.071
18.5 ≤ a < 25.0	2224 (70%)	56 (72%)	
a < 18.5	297 (9%)	12 (16%)	
Exercise habit ^b^	2009 (63%)	44 (56%)	0.221
Malignant tumor ^b^	522 (16%)	14 (18%)	0.718
Diabetes ^b^	1208 (38%)	33 (42%)	0.438
Dyslipidemia ^b^	1852 (58%)	41 (53%)	0.316
Musculoskeletal disorders ^b^	2458 (77%)	61 (78%)	0.850
Circulatory disease ^b^	1700 (54%)	58 (74%)	<0.001 *
Support/care-need certification ^b^	450 (14%)	32 (41%)	<0.001 *
Medicine taken regularly ^b^	2794 (88%)	70 (90%)	0.615
Number of present teeth (tooth)			
-19	1042 (33%)	39 (50%)	0.001 *
20-	2138 (67%)	49 (50%)	
Periodontal pockets (mm)			
<4	876 (28%)	29 (38%)	0.054
≥4	2292 (72%)	48 (62%)	
Tongue and lip function ^c^	970 (31%)	39 (50%)	<0.001 *
Swallowing function ^c^	375 (12%)	24 (31%)	<0.001 *

Abbreviation: BMI, body mass index. * *p* ≤ 0.05 using Fisher’s exact test or the Mann–Whitney *U* test. ^a^ Male (proportion of males); ^b^ Presence (proportion of presence); ^c^ Poor (proportion of poor status).

**Table 2 healthcare-12-02197-t002:** Comparison of serum albumin levels and swallowing function.

Factor	Serum Albumin Level			*p-*Value
	a < 3.6 G/dL (*n* = 78)	3.6 G/dL ≤ a < 3.9 G/dL (*n* = 339)	3.9 G/dL ≤ a (*n* = 2841)	
Swallowing function				
Good	54 (69%)	289 (85%)	2516 (89%)	<0.001 *
Poor	24 (31%)	50 (15%)	325 (11%)	

* *p* ≤ 0.05 using Fisher’s exact test. a: serum albumin level.

**Table 3 healthcare-12-02197-t003:** Crude OR and 95% CI for a low serum albumin level.

Factor		OR	95% CI	*p-*Value
Age (years)		1.169	1.124–1.216	<0.001
Gender	Female	1	(reference)	0.032
	Male	1.636	1.043–2.566	
BMI (kg/m^2^)	≥18.5	1	(reference)	0.076
	<18.5	1.765	0.943–3.302	
Exercise habit	Absence	1	(reference)	0.223
	Presence	0.754	0.479–1.187	
Malignant tumor	Absence	1	(reference)	0.718
	Presence	1.114	0.620–2.001	
Diabetes	Absence	1	(reference)	0.438
	Presence	1.197	0.790–1.887	
Dyslipidemia	Absence	1	(reference)	0.317
	Presence	0.795	0.507–1.246	
Musculoskeletal disorders	Absence	1	(reference)	0.850
	Presence	1.054	0.612–1.816	
Circulatory disease	Absence	1	(reference)	<0.001
	Presence	2.525	1.512–4.217	
Support/care-need certification	Absence	1	(reference)	<0.001
	Presence	4.220	2.659–6.699	
Medicine taken regularly	Presence	1	(reference)	0.615
	Absence	1.209	0.577–2.532	
Number of present teeth (tooth)	20-	1	(reference)	0.001
	-19	2.052	1.308–3.218	
Periodontal pockets (mm)	<4	1	(reference)	0.056
	≥4	0.634	0.397–1.012	
Tongue and lip function	Good	1	(reference)	<0.001
	Poor	2.278	1.452–3.574	
Swallowing function	Good	1	(reference)	<0.001
	Poor	3.324	2.031–5.442	

Abbreviations: OR, odds ratio; CI, confidence interval; BMI, body mass index.

**Table 4 healthcare-12-02197-t004:** Adjusted OR and 95% CI for low serum albumin level.

Factor		OR	95% CI	*p-*Value
Age (years)		1.115	1.064–1.169	<0.001
Gender	Female	1	(reference)	0.001
	Male	2.208	1.360–3.584	
Circulatory disease	Absence	1	(reference)	0.025
	Presence	1.829	1.079–3.099	
Support/care-need certification	Absence	1	(reference)	0.008
	Presence	2.087	1.208–3.606	
Number of present teeth (tooth)	20-	1	(reference)	0.416
	-19	1.223	0.753–1.984	
Tongue and lip function	Good	1	(reference)	0.260
	Poor	1.323	0.813–2.155	
Swallowing function	Good	1	(reference)	0.002
	Poor	2.379	1.377–4.112	

Abbreviations: OR, odds ratio; CI, confidence interval. Adjusted for age, gender, circulatory disease, support/care-need certification, number of present teeth, tongue and lip function, and swallowing function. Hosmer–Lemeshow test: *p*-value = 0.136.

## Data Availability

The data presented in this study are available on request from the corresponding author. The data are not publicly available due to ethical restrictions.

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
