# Peer review of "Relationship Between Swallowing Function and Low Serum Albumin Levels in Older Japanese People Aged ≥ 75 Years"

_healthcare, 2024, doi:10.3390/healthcare12212197_

Round 1

Reviewer 1 Report

Comments and Suggestions for Authors

There are some revisions that I would like the authors to address and/or to consider.

1.      Abstract should be formatted according to the journal format

2.      Abstract conclusion needs to be revised. Current statement is a repetitive of result.

3.      As data is a large data Kolmogorov–Smirnov test is not appropriate for testing normality.

4.      Please amend : All p-values < 0.05 to   p-values ≤ 0.05

5.      Please discuss in discussion section:  Positive cases (serum albumin level ³ 3.6 G/d)  are only 78 cases , this issue substantially affect the results. Results are statistically significant, but they are negligible.

6.      Please omit the paragraph “In this study, the Hosmer–Lemeshow test was used for the multivariate logistic re- gression analysis model. The Hosmer–Lemeshow test is used to examine the goodness- of-fit of multivariate logistic regression analysis models, testing whether the observed event rates in a subgroup model fit the expected event rates. The Hosmer–Lemeshow test is considered to indicate a good fit with p-values > 0.05 [55]. In this study, the p-value was0.136, suggesting a good fit” from discussion and move to the Statistical analysis section

7.      Please provide clinical and theoretical implications of the study

Reviewer 2 Report

Comments and Suggestions for Authors

In their manuscript "Relationship between Swallowing Function and Serum Albumin Levels in Japanese Older People Aged >= 75 years" the authors investigate the relationship between nutritional status, dependence (as an issue that negatively affects nutritional status) and swallowing function.

I will focus on some general considerations that emerged after reading the manuscript.

Albumin is already an indicator of nutritional status in itself. In recent years it was highly contested, even abandoned, and only recently revived in Europe but to be used with other variables. Therefore, this indicator's usefulness should be well highlighted and explained together with its limits in detecting the nutritional status of a subject.

The introduction is very convoluted. The authors talk about the risks that a nutritional status entails if compromised, about how this leads to dependence, and how the dependence in turn leads to malnutrition. Dysphagia is presented as one of the potential causes of malnutrition which actually it is. It seems to me that the central point (the dependent variable) should be swallowing function and not albumin! Analyzing the data in this way would be much more useful, in my opinion.

The authors could show how much the albumin levels and BMI differ (or do not differ) in subjects with swallowing problems vs. subjects without difficulties with swallowing. Then one could hypothesize a certain usefulness of albumin in detecting the impairment of nutritional status through the impairment of swallowing function.

The title of the manuscript should be "Can serum albumin be a valid/useful indicator of nutritional risk and need for care in subjects with swallowing problems" or something like that.

Other considerations: English must be revised in some parts, the considerations on the ethics committee should be placed at the beginning of the paragraph on the methodology, the inclusion and exclusion criteria should be presented, and a flow chart that explains the various phases of the study should be included. The authors do not mention that 2 different cut-offs for albumin will be used in the methodology section. There is no need to explain what statistical tests are used for each variable specifically; it should be only explained for different categories of variables (continuous and categorical). In results: when explaining tables just the name of the variable “serum albumin”, “support/care need certificate”, “teeth” without the term “presence” before.

Round 2

Reviewer 1 Report

Comments and Suggestions for Authors

I would like to thank the authors for considering the comments and changing the manuscript accordingly.